# Estimating the effect of COVID-19 on mental health: Linguistic indicators of depression during a global pandemic

**JT Wolohan**
Booz Allen Hamilton
wolohan_john@bah.com

## Abstract

This preliminary analysis uses a deep LSTM neural network with fastText embeddings to predict population rates of depression on Reddit in order to estimate the effect of COVID-19 on mental health. We find that year over year, depression rates on Reddit are up 50% , suggesting a 15-million person increase in the number of depressed Americans and a $7.5 billion increase in depression related spending. This finding comes at a time when uncertainty about the impact of COVID-19 on physical and economic health is still high, and suggests that in addition to those factors, mental health must be considered as well. As data becomes available, further research will be needed to validate the results of this preliminary investigation.

## 1 Introduction

The COVID-19 pandemic has already plagued the people of the world's physical health, and while the impact of the novel coronavirus on our mental health is less well understood, it is expected to be negative (Wang et al., 2020; Ammerman et al., 2020). Even those who never get sick during a pandemic can experience a multitude of psychological stressors during a disease outbreak and those stressors can persist well past the end of the outbreak (Chew et al., 2020). The popular media is aware of the necessity for otherwise healthy people to emphasize "self care"—small acts intended to maintain one's mental health or relieve stress—during these uncertain times. This analysis attempts to quantify the impact that COVID-19 has had to date on the population rate of depression through the use of state-of-science depression prediction models and data from the popular social media site Reddit.

This paper continues an established line of research in the application of natural language processing techniques to the disease of depression (Guntuku et al., 2017), and mixes it with the rapidly emerging field of COVID-19 research (McKibbin and Fernando, 2020; Duan and Zhu, 2020). Research in the former area is centered around the notion that language use reflects the thought processes of the speaker and that by assessing the words people use, we can gain insight into their thought processes (Fine, 2006). From this, it follows that text classification approaches such as the use of long short-term memory networks (Hochreiter and Schmidhuber, 1997) and word embeddings (Mikolov et al., 2013), such as fastText (Bojanowski et al., 2017), can be used to classify people's mental health status based on their speech. Depression has been studied widely in this way due to its grave impact on those it afflicts (De Choudhury et al., 2013; Coppersmith et al., 2015). Indeed, even subclinical levels of depression have been shown to reduce quality of life in meaningful and measurable ways (Cuijpers and Smit, 2002).

## 2 Method

In this paper, we use data from Reddit to measure the potential impact of COVID-19 on depression. We do this in two parts: first, we extend the approach from Wolohan et al. (2018), using an LSTM model to improve the accuracy of depression prediction on social media; then, with this model, we analyze a new dataset—composed of the Reddit comments of 20,000 users across the first six months of 2018, 2019, and 2020[1]—in order to estimate the population rate of depression during the COVID-19 pandemic.

### 2.1 Data

For these analyses we use two datasets of Reddit comments, aggregated at the user level. The first dataset–the Off-Topic Depression dataset–comes from Wolohan et al.; The second, is a novel dataset

---

[1]This paper was written in April, 2020. More data will be gathered as it becomes available.

created for this task: the Reddit Pandemic Depression dataset.

The Off-Topic Depression dataset contains 141-million words from Reddit comments, aggregated by 11,000 users. The text in this dataset was not allowed to come from subreddits—the site-wide term for a themed message boards—where discussion of depression or a related issues was expected. Therefore, this dataset contains only "off-topic" text—text not on the subject of depression. This step is important for being able to detect depression among the general public, many of whom are reluctant to talk about their depressive symptoms due to depression-related stigma (Manos et al., 2009). We label users as either "depressed" or "not-depressed" based on self-disclosure behavior: authoring posts in depression-related subeditors. The Off-Topic Depression dataset is useful for training a model because of the available baseline, but contains no data from the recent COVID-19 pandemic. This makes it insufficient to estimate the impact of COVID-19 on population rates of depression.

### 2.1.1 Pandemic Depression dataset

The Pandemic Depression dataset contains 23 million words generated by 20,000 users over three years. The users for this dataset were selected in a similar fashion to the users from the Off-Topic Depression dataset: a scrape of submission authors to the subreddit r/AskRedit was performed to gather a set of potential users and then a random subset from this set was taken. During the scrape, 29-million users were considered. From that sample, 20,000 were randomly selected for inclusion in the study. This approach attempts to gather "neutral" Reddit users, who are not necessarily associated with any particular subreddit or community of subreddits, and would therefore have no bias towards or away from depression. The data for the Pandemic Depression dataset is broken up by the time of users activities. We use only the first six months of 2018 and 2019 and the first four months of 2020[2].

### 2.2 Deep LSTM with fastText

As part of this analysis we trained a deep long short-term memory neural network. The network we used contains five layers: a fastText (Joulin

| Month | 2018 | 2019 | 2020 |
|-------|------|------|------|
| Jan | 1.6 mil | 5.5 mil. | 560k |
| Feb | 475k | 1.8 mil. | 860k |
| Mar | 335k | 1.2 mil | 2.4 mil |
| Apr | 258k | 1 mil | 5.6 mil |
| May | 211k | 700k | * |
| Jun | 210k | 470k | * |
| * Data not yet available. | | | |

Table 1: Pandemic Depression dataset text by month.

et al., 2016) embedding layer, three LSTM layers, and an output layer. We trained and evaluated the LSTM on the Off-Topic Depression dataset, using about 7,700 users for training and about 4,000 users for testing. This totalled about 100 million words for training and 40 million for testing.

The first layer of the model, the embedding layer learns weights that take advantage of the specific 300-dimensional fastText vectors. The second through fourth layers of the model are identical LSTM layers with a 20% dropout rate. The LSTM layers use each word as a step. The fifth and final layer of the model is a single-node dense layer with sigmoid activation used for predicting the class of the user. A depiction of this network and the minimal prepossessing can be seen in Figure 1.

Text preprocessing for the LSTM was minimal. The vocabulary for all documents was limited to the most-common 10,000 words. Each user was truncated down to or zero-padded up to 750 words, as necessary. We performed no other preprocessing, such as misspelling correction or internet-speak normalization.

### 2.3 Comparative time-series analysis

In order to assess the impact of the COVID-19 pandemic on language use, we perform a comparative time-series analysis of three periods: two six month periods from before the pandemic ranging from January 2018 and 2019 to June 2018 and 2019 inclusive, and one four month period from January 2020 to April 2020 [3]. We analyzed the same users for all periods. Rates of depression were estimated for each period by classifying each user as either depressed or not-depressed with the LSTM classifier described in section 2.2.

It is important to note here that Reddit activity is inconsistent and that, unlike other social media

---

[2]Historical Gaffney and Matias (2018) note that there are issues with historical analysis of Reddit data—perhaps inclusive of users deleting depressive content post-hoc. Addressing those concerns are outside the scope of this preliminary investigation.

[3]Data from April was only available up to April 4; however data is included because a large enough volume of data was accessible: $\approx 50,000$ words.

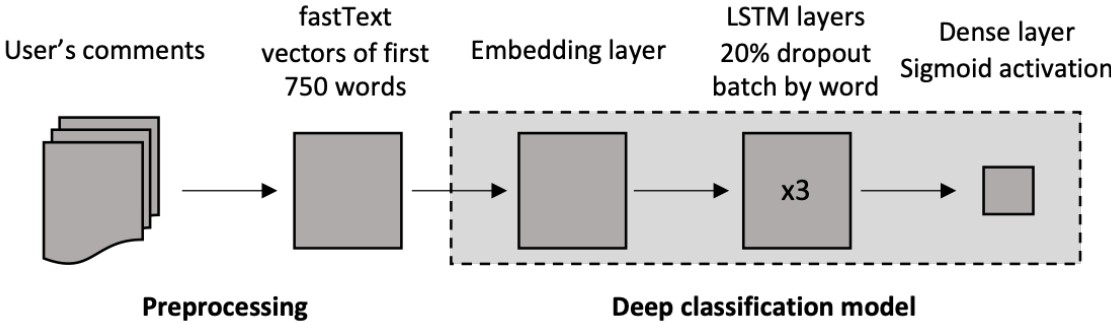

Figure 1: Deep LSTM model for depression prediction.

platforms such as Twitter or Facebook, the use of a single account through time is not encouraged by the platform (Leavitt, 2015). This results in many accounts being abandoned and the resulting 2020 subset of data being smaller in terms of total active users than the subset of data for 2019. A more appropriate means of performing this analysis would be to select a random sample users known to be active during this time period in 2020. Requiring the same users for all periods may introduce bias if users who are likely to be active over long periods of time have a bias towards or away from depression.

## 3 Preliminary results

In this section, we review the preliminary results. We find that an LSTM with fastText embeddings outperforms the baseline approach in Wolohan et al. Additionally, the LSTM indicates that the population rate of depression may be up by 50% in the first four months of 2020 when compared to the first four months of 2019 and 2018.

### 3.1 LSTM with fastText embeddings

Comparing the new model for off-topic depression prediction, a deep LSTM with fastText word embeddings, to the model previous used by Wolohan et al., We find that the LSTM outperforms the previous baseline approach in the relevant measures of AUC and F1 score. The LSTM achieved an AUC of 0.93 and an F1 score of 0.92, surpassing the baseline by 18 points 24 points respectively. The results for this LSTM are competitive with state-of-the-art deep-learning approaches for this task on similar datasets (see: Orabi et al. 2018; Guntuku et al. 2017).

| Model | AUC | F1 |
|---|---|---|
| Wolohan et al. | 0.75 | 0.68 |
| LSTM + fastText | 0.93 | 0.92 |

Table 2: LSTM performance versus baseline.

### 3.2 Comparative time-series analysis

With the LSTM model improved to 0.93 AUC, we then applied the LSTM to the Pandemic Depression dataset. In doing this, we assessed whether or not users' language indicated depression for the first six months in 2018 and 2019, and the first four months of 2020. We found for 2018 and 2019, the user population being studied demonstrated a steady rate of depression around 33%±4%. For 2020, we found the population rate of depression to average 49%, with individual months ranging from 42% to 52%.

In the first six months of 2018 and 2019, the LSTM suggested a depression rate in the low 30% range with only two exceptions: May 2018 and January 2019. May 2018 had the highest estimated depression rate–38% of all users–while January 2019 had the lowest–29% of all users .

In 2020, estimated depression rates among Reddit users again are consistent; however, they are consistently 20 percentage points higher than 2018 and 2019. Of the three months, only April 2020 stands out with a low depression rate: 42%. At the time of this writing, the data for April 2020 is incomplete.

## 4 Discussion

With the global COVID-19 pandemic wrecking havoc on medical systems and economies worldwide, 2020 will be a year in which many people go through significant hardships. If the analysis herein is to believed, then the fear that many have about a

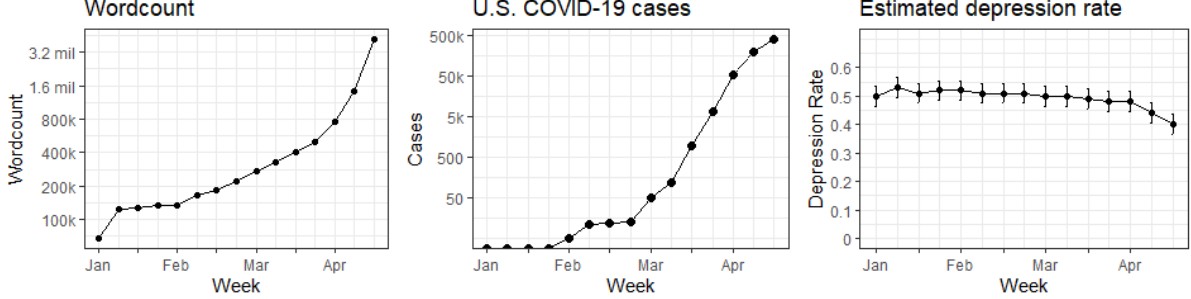

Figure 2: Sampled word count, COVID-19 cases, and modeled depression rate by week.

| Month | 2018 | 2019 | 2020 | Δ |
|---|---|---|---|---|
| January | .32 | .29 | .52 | +79% |
| February | .34 | .32 | .51 | +69% |
| March | .31 | .32 | .49 | +53% |
| April | .33 | .34 | .42 | +24% |
| May | .38 | .32 | - | - |
| June | .34 | .34 | - | - |
| Average | .34 | .32 | .49 | +53% |

Table 3: Estimated depression rate of Reddit users for select months.

global deterioration in mental health is likely to be a reality as well. But should we believe the analysis presented here; and what are the implications if we do?

### 4.1 Model efficacy

There are two reasons that we should tentatively believe the AI-based assessments of population-level depression on Reddit. First, the 32% population-rate of depression estimated by the model is plausible, given (1) that the LSTM is designed to detect both clinical and subclinical depression and (2) that Reddit has a much younger (read: depression prone) population than the U.S. at-large. Second, the steadiness of the numbers over time is encouraging.

First among the reasons that one would avoid dismissing these findings out of hand is the consistency between the numbers projected for 2018 and 2019 and what one would expect for a population joint rate of clinical and subclinical depression on Reddit. According to the U.S. National Institute of Health, adult rates of depression range up to 13% depending on the demographic[4]. At the upper bound of 13% is the 18-25 year-old demographic.

More than half of the Reddit-using population falls into this demographic[5]. Further, 25% of Reddit users are 17 or younger—suggesting they might also have an increased rate of depression. NIH estimates that approximately 17% of adolescents between 15 and 17 will experience at least one major depressive episode each year. If one takes 13% as the population rate of depression for Reddit users and doubles it to include cases of subclinical depression (Kessler et al., 1997), one is left with a 26% rate of total depression for Reddit users for the first six months 2019. The LSTM predicts a 32% rate of depression for the first six months of 2019. A reasonable amount of error, given the accuracy measures in Table 2.

Second, the steadiness of the rates of depression in the two control years is encouraging. There is little variation across the first six months of both 2018 and 2019. With only 9 percentage points separating the month with the greatest estimated rate of depression, May '18, and the lowest month, Jan. '19.

### 4.2 Estimating the effect of the COVID-19 pandemic on mental health

If we assume that the rates of population-wide clinical and subclinical depression are to increase by 50% in 2020, either as a result of the COVID-19 pandemic or otherwise, then we would expect to see a population rate of depression increase from 7% to 10%, and a similar sized increase sub-clinical depression. In the U.S., this would amount to 15 million more adults suffering from clinical depression. This increased rate of depression would amount to a $7.5 billion increase in healthcare spending, assuming $500 per person per year (Kleine-Budde et al., 2013). This assumes that

---

[4]All population rates of depression come from the NIH: https://www.nimh.nih.gov/health/statistics/major-depression.shtml

[5]There are two sources for the demographics of Reddit, both from 2016: Barthel et al. (2016) or /u/HurricaneXriks (2016). The latter is used here.

the average rate of depression is about as severe as it is currently.

Importantly, there reasons we may believe this increased rate is not yet associated with COVID-19. As we can see in Figure 2, the estimated depression rate appears to be declining in April at a time when cases and deaths in the U.S. are rising. Speculatively, this may be associated with the phase of the pandemic the U.S.—and therefore most Reddit users—are currently experiencing. Many of these users will be under stay-at-home orders; however, the full toll of the pandemic, including economic destruction and loss of life has not yet been felt. Alternatively, we must consider that the active population of Reddit may be changing as stay-at-home orders and unemployment furnish people with addition free time to use the internet.

One would expect that depression rates increase as stressors such as unemployment, loneliness, and loss of loved ones begin to impact the population at large. If the population rate of depression is elevated for a reason un-related to COVID-19, that could spell even further trouble for Americans' mental health.

## 5 Ethical considerations

As with all works related to public health and mental health, it is important that we consider the ethical implications of our research and make explicit the ethical justification for the work. In particular, work of this kind–namely, public health surveillance–requires special attention because, while disease surveillance is foundational to good public health practice (Fairchild et al., 2007), it also forgoes the notion of informed consent. When discussing a health issue rife with stigma such as depression, the concern about a researcher mismanaging data and revealing public health information of individuals without their consent is amplified.

Klingler et al. (2017) enumerate eight broad categories of ethical arguments by which researchers justify forgoing the traditional informed consent requirement for conducting public health surveillance. Of those, we argue that our work satisfies the effectiveness, necessity, proportionality, and least infringement requirements. First, with respect to the effectiveness, this study is the first–to my knowledge–quantitative estimate of the impact of COVID-19 on population rate of depression. That makes this data valuable to the public health and mental health communities. Second, with respect

to necessity, it is noteworthy that this approach is minimally intrusive, requiring no contact with the individuals whose comments are used and no interruption of their usage of the Reddit service. For proportionality, third, it is important to note that no individual user-level data was shared at any time during this research, and that the identities of Reddit users are hidden behind pseudonyms, offering them an additional layer of protection. Fourth and finally, the work considered carefully the notion of least infringement, collecting only data that would be necessary for the analysis herein.

Additionally, in a departure from traditional practices in the NLP community, the data underlying this work will only be shared with researchers who both (1) provide a research design or other public-health justification for the use of the data and (2) agree to take the necessary efforts to secure the data.

Ultimately, we view this work as ethically justified based on the precautions noted above and the potentially large increase in population-level depression against which this research warns. If depression is, as we predict, to impact 15 million more Americans through 2020 than in previous years, advanced warning is valuable to the American mental-health system.

## 6 Conclusion

In this paper we show the effectiveness of an LSTM text classifier using fastText word embeddings at predicting user-level depression and use that classifier to estimate the population rate of depression in April 2020 in the midst of the COVID-19 pandemic. We estimate that through the first six months of 2020, population rate of depression is up $\approx 50\%$, corresponding to a 15 million more depressed Americans. This analysis suffers from a lack of data and will strengthen as more data becomes available. Additional research is needed to confirm or contradict the results presented here, and will be especially valuable when the adjusted population rates of depression are known.

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
