# OpenReview forum: "Estimating the effect of COVID-19 on mental health: Linguistic indicators of depression during a global pandemic"
_aclweb.org/ACL/2020/Workshop/NLP-COVID — NLP-COVID-2020_

### Official Review · AnonReviewer1 · 2020-04-20
**Beautifully written paper**

**Rating:** 10
**Confidence:** 4

**Review:**

This might be the most beautifully-written paper I have seen in a fairly long career.  With the removal of two lines in the Conclusion section, I would happily urge immediate publication, and would use it as an example in writing courses.

STRENGTHS:

Beautifully written Introduction that clearly establishes the “real-world” (i.e. non-natural-language-processing) motivation for the work with appropriate citations to the clinical psychology literature. It then goes on to make a clear statement about the goals of the paper, and makes a reasoned argument for the innovation of the work (rather implicitly, but still, it’s there) with respect to the natural language processing literature. So: clear statement about what the paper is going to do, well-argued significance claim, and well-supported innovation claim.

The construction of the data set is really nicely done, especially with the use of only the first 6 months of data from each year.  This presumably is meant to assume that when later looking at the Pandemic Depression dataset, there is no confound from seasonal effects on mental health (the pandemic having entered broad public consciousness in the US in the early months of 2020, and this being only April--I mention this because the pandemic dates back to at least November of 2019 and there was widespread awareness of this at the highest levels of the US government well before that entrance into broad public consciousness, which is certainly relevant to discussions of the materials, since it speaks directly to the appropriate months to include).  Consider making that seasonal effect explicit, if that is, indeed, the intent--it’s a strong point of the methodology.

The discussion of the characteristics of the dataset in the second paragraph of Section 2.3 (“It is important to note here…”) gives excellent insight and background.  Additionally, I would point out that it is the paper’s adroit use of discourse markers like “It is important to note here” that makes this paper such an excellent example of scientific writing.

The presentation of potential alternative explanations for the findings in the Discussion section is one of the most beautiful parts of the paper.  Respectful consideration of alternative points of view increases the reader’s confidence in the overall reasoning behind the paper immensely--very nicely done.

WEAKNESSES

The paper gives very few details about the LSTM that is at the heart of the methodology, and very detail about preprocessing, as well.  Especially given that there does not seem to have been much of the latter and so it presumably wouldn’t take up much time or space, consider adding those details to an appendix.

The two lines that I would strongly recommend removing: “In this paper we show the effectiveness of an LSTM text classifier using fastText word embeddings at…” The paper actually does not show that particularly well.  The only point of comparison FOR THE PERFORMANCE OF THE LSTM/EMBEDDINGS COMBINATION ITSELF is to previously published results. When making claims based solely on performance numbers, the level of detail available to the reader who wants to reimplement the work becomes absolutely crucial. As Kenneth Church has put it: “The better the numbers are, the more important it is to reject the paper. We can't afford papers that report results without insights.” ...and there are no insights REGARDING THE LSTM/EMBEDDINGS COMBINATION that I was able to find in this paper.  Where would they come from? Manipulation experiments--swapping in and out sources of embeddings, swapping in and out other classifiers, etc.  Note that this is an issue of what can and cannot be said about the NLP--it does not take anything away from the beautifully discussed findings regarding depression.

In contrast to that 2-line conclusion regarding the LSTM/embeddings: the analysis of the results in terms of comparison with (a) reported data on depression rates, and (b) on the variability or lack thereof in the two control years--is really creative and insightful.  See all 3 paragraphs on Section 4.1, Model Efficacy, for that analysis.

So: to strengthen the paper, consider making it not about how well the LSTM/embeddings combination performs, but rather about the findings regarding depression.  We are talking here about removing two lines that make the paper’s conclusions be fully supported by the methodology and its results.

Again: overall, this paper is exemplary. Removal of the two lines identified above would make it even better.

---

### Official Review · AnonReviewer3 · 2020-04-23
**Interesting and useful conclusion, but the method and analysis could improve.**

**Rating:** 7
**Confidence:** 3

**Review:**

This paper conducts an analysis of the depression rate of users posting in reddit before and after 2020. The results show a sharp increase of depression in 2020, which coincides with the start of COVID-19. The paper is worth publishing in that it shows a sharp increase of depression and this needs to be addressed.

The paper concludes that COVID-19 caused the increase of depression but the analysis conducted in the paper is not clear, only that there is a correlation.

The method used is OK and the system shows good results when evaluated on the development data, but it has some parts that can be improved, especially if the absolute numbers of depression reported are to be accurate, they might not be. But the increase of depression is clear based on their experiments.

Additional comments:

The system was trained on a separate training data using reddit, and then applied the system to the new collection of reddit posts. It was not clear to me whether there was an overlap of users/posts between the training and test data. The paper must say what they did to prevent/control this overlap. Still, this is just a minor comment. If there was any overlap, then the depression results in 2018 would be artificially high. Given that they are similar to those of 2019, it seems that there is little or no overlap between training and test data.

The paper gives a detailed explanation of the plausibility of such a high rate of depression in all years. What they say makes sense, but I noted that the training data has an artificially high number users with depression (from the cited paper: 4000 depressed vs 7000 control). It would be best to conduct some simple error analysis. For example, the system might have a high rate of false positives?

The highest value of depression found was in January 2020. If COVID-19 was the cause, why does the rate of depression declines slightly during 2020? Can the paper report on the geographical profile of the reddit users? If they are mostly Chinese, the high increase of depression in January would make sense. If they are mostly from USA, why isn't there higher depression in March? There could be confounding factors here.

Something to consider for further work: analyse the topics discussed in the posts, or at least analyse the frequency of covid-related posts among depressed and non-depressed users. This can help establish whether there is causation.

Section 2.3 mentions that they analysed the same users for all periods but I do not see anywhere any analysis of how many users changed from not depression to depression (or otherwise). Is this possible?

Some small comments/typos below:

- Abstract: near the middle of the abstract, there is a line starting with a comma.

- Page 1, introduction: "for otherwise healthy emphasize" -> "for otherwise healthy to emphasize"

- Page 1, section 2.1: The reference should include the year 2018.

- Page 4, beginning of col 2: "Importantly, there are reasons"

---

> ### Comment · AnonReviewer3 · 2020-05-06
> **Agree to expect more detail of LSTM system for reproducibility**
>
> I agree with the other reviewers that it would be best to give more detail of the LSTM system used. I am comfortable with referring previous work for the details of the data.
>
> A reason why my recommendation is accept despite some unclear details in method an analysis is that this paper is timely and potentially useful. I feel that the lack of details of the LSTM system is important for reproducibility only, and some short description and/or link to a software repository will suffice.
>
> If the aim of NLP-COVID19 is to publish only completed work I'll be happy to reduce the final score to rating 6.

---

### Official Review · AnonReviewer2 · 2020-05-02
**Worthwhile work and interesting discussion but a lot of technical detail missing.**

**Rating:** 4
**Confidence:** 5

**Review:**

This paper is about estimating the effect of COVID-19 on depression by using an LSTM to calculate depression from the language in Reddit users' posts at different periods prior to and during the pandemic and making comparisons. The paper has an interesting goal and I like the fact that it argues for a longitudinal analysis. However a lot of technical details are missing, including the task definition and the associated ground truth, and I would find it difficult to accept the paper in its current form.

Major comments
--------------
At the moment this paper relies too heavily on information from a previous publication by the author from 2018. However, it should be possible to read the current paper as a standalone. Therefore I suggest that the following information is included here:
Some more details about the off-topic depression dataset. What was the criterion for selection users and posts in this dataset and how was it annotated for depression? While I agree that most people posting on social media will not be explicit about depression or sad mood it is important to know how the ground truth for depression was established in the off-topic dataset. What is the available baseline being mentioned just before section 2.1.1?

Method: The paper doesn't actually describe anywhere the actual task that the LSTM is trained on and this is even more confusing as we don't know what the ground truth is and at what level it is assumed.

Deep LSTM with fastText: it is totally unclear what is the task here. LSTM is trained on some users using what as the ground truth? What is the end goal? Is it the binary classification of a user as depressed or not depressed? Also, what is the input to the LSTM? Is a single post considered to be one time-step? Or is each word one time-stephere? Very little technical detail is provided and clarifications are needed before this work can be published.

Pandemic depression dataset:
Before the random selection of 20K users were there any other filtering criteria for candidate users (e.g. minimum threshold on number of posts etc). Are only the first six months of 2018, 2019 to make any seasonal effects as close to 2020 as possible?

Section 3: What was the baseline by Wolohan et al 2018, also mentioned in Table 2?
How are the rates of depression determined?  Is this by aggregation the number of depressed users in a time period of the percentage of posts indicating depression?

The discussion is interesting but would be more useful if it was made clear what the prediction task was, what the input to the model was and how this information was aggregated to obtain rates of depression.

Minor comments:
-----------------
The following sentences are ungrammatical and needs rephrasing:
"The popular media is aware
of the necessity for otherwise healthy emphasize
“self care”—small acts intended to maintain one’s
mental health or relieve stress—during these uncertain
times."

"The Pandemic Depression dataset contains 23 million
words aggregated up to 20,000 users(...)"

The following could be shortened or removed if more space needs to be found:
"From this, it follows
that text classification approaches such as the
use of long short-term memory networks (Hochreiter
and Schmidhuber, 1997) and word embeddings
(Mikolov et al., 2013), such as fastText (Bojanowski
et al., 2017), can be used to classify people’s
mental health status based on their speech."

Section 4.2 can also be shortened to make more space for necessary technical detail in earlier sections.

---

> ### Comment · AnonReviewer2 · 2020-05-06
> **Agree that conclusions are important and very much in favour of longitudinal analysis. But we also need to understand the task..**
>
> I would also like to accept this paper for a very important conclusion and performing a longitudinal analysis.
>
> However, I am facing a dilemma as I strongly believe that omitting the task definition and important aspects of the methodology are not minor points.
> What is the task? Is it the binary classification of a user as depressed or not depressed? How is this information aggregated to obtain depression rates? What data is the model trained on (all data from specific users in a particular time-span)? How was the ground truth obtained? The connection with the 2018 paper is not clear and even there it is not obvious how the off-topic ground truth for depression was obtained. What is the input to the LSTM? Is a single post considered to be one time-step? Or is each word one time-step here? What is the off-topic baseline mentioned from the 2018 paper?
>
> Being clear about the task is crucial for reproducibility but also in order to be able to sanity check any results. While my understanding is that the purpose of this workshop is to showcase NLP work in the context of COVID-19, I don't think we should be bypassing the need for methodological clarity. This is still an NLP venue.

---

### Comment · Program_Chairs · 2020-05-14
**Revise and Resubmit**

There was some disparity between the reviews for this paper. Taking into consideration this feedback, we recommend that the author submits a revision of the paper that includes the information regarding the task definition and other methodological clarifications as per the comments from Reviewer 2. This will address the key concerns of the reviewers.

Thank you for your interesting submission which has clearly sparked important discussion! We look forward to the revision.

---

### Decision · Program_Chairs · 2020-10-15

**Decision:**

Accept

**Comment:**

Entering decision previously communicated after revision.